# If I were you Minority and majority members evaluate relevancy and subjective experience differently while putting themselves in the other's shoes

Dalit Milshtein[1]*, Ahmad Serhan[2], Simone Shamay-Tsoory[2]

**1** The Berlin School of Mind and Brain, Humboldt-Universität zu Berlin, Berlin, Germany, **2** Department of Psychology, University of Haifa, Israel

* dm.in.humboldt@gmail.com

## Abstract

Previous research suggests that asymmetrical power relations between minority and majority groups manifest in various cognitive biases. However, the underlying processes driving these differences remain unclear. Here we examine whether minority and majority groups exhibit differences in their basic ability to imagine outgroup-related scenarios. In two experiments, we asked participants from the majority (Israeli Jews) and minority (Israeli Arabs) groups to put themselves in their respective outgroups' shoes. The results indicated that majority members perceived outgroup-related themes as less imaginable, relevant, and plausible to themselves compared to the minority group. These findings suggest that an individual's social power and hierarchical position within society may influence their ability to imagine themselves and others, subsequently contributing to intergroup bias.

## 1. Introduction

### 1.1. Intergroup bias, general concept

The capacity for creating and maintaining social systems lies deep within our human evolutionary heritage [1]. While group living allows for cooperation and improved access to resources, it also introduces the potential for conflicts and challenges in intergroup relations, such as *intergroup bias* [2,3]. Intergroup bias may cause one to judge, comprehend, and interpret 'Others' in a manner that transcends the objective requirements or evidence of the situation, thereby creating a psychological blind spot [4]. Intergroup bias has been implicated as the likely psychological foundation of prejudice, stereotyping, and discrimination and is evident in various behavioral phenomena, such as ingroup favoritism or outgroup hostility [5,6]. Given that the early twenty-first century is characterized by growing nationalist sentiment and increasing global migration, intergroup relations have far-reaching implications across multiple domains [7].

**Data availability statement:** All data and code used for stimulus presentation and analysis are publicly available on the OSF Repository at: https://osf.io/ufxzn.

**Funding:** The author(s) received no specific funding for this work.

**Competing interests:** The authors have declared that no competing interests exist.

Groups differ from each other not only in internal attributes such as language, culture, and religion but also in terms of resource availability, political or social power, and status. These human group-based hierarchies result in asymmetrical intergroup relationships [6,8]. Asymmetrical power relations between diverse groups can give rise to varying levels of intergroup bias, leading groups to exhibit varying susceptibilities to such biases. This is particularly relevant when examining the dynamics between minority and majority groups [e.g., 9, 10]. Four leading theories offer different frameworks for inspecting intergroup dynamics. *Social identity theory* (SIT) [6, 11] focuses on the social categorization process and its effect on the motivation to perceive ingroups more positively than outgroups. *Realistic group conflict theory* (RGCT) [12,13] focuses on the conflicting interests of the parties in question, which may involve competition over limited resources. Building on RGCT and SIT, *Intergroup Threat Theory* (ITT) [14], explains bias in group relationships by examining the perception of outgroups as threats. The *Social Dominance Theory* (SDT) posits that asymmetrical power dynamics result in behavioral asymmetry, with both dominant and subdominant groups contributing equally [8].

## 1.2. Perspective-taking as a fundamental component in intergroup bias

Despite the vast body of research, the specific cognitive mechanisms behind ingroup biases remain mostly unknown. Nevertheless, the ability to put oneself in others shoes was suggested as one of the main factors that may play a substantial role in intergroup relations [15–17]. This ability has been conceptualized as perspective taking, a contextual process for understanding another person's mental state. It is supported by a social cognitive capability that develops over time and varies as an individual trait [18]. Specifically, previous studies suggest that enhanced perspective-taking ability correlates with reduced intergroup bias and increased pro-social behavior. For example, children with greater perspective-taking skills were also more inclined to behave in a prosocial way during peer interactions [19]; perspective-taking has been found to effectively reduce intergroup conflict [20], and observing that another person shares one's perspective leads to heightened affection and higher assistance offered to that individual [21]. On the other hand, lack of perspective-taking has been linked to aggressive behavior [e.g., 22, 23]. Considering the aforementioned, exploring the ability of minority and majority groups to take perspective of each other may provide us a better understanding of the core components that compose the intergroup relations.

## 1.3. The perspective-taking process components

Perspective-taking can be greatly influenced by the cognitive and emotive components involved while formulating alternative perspectives or states of affairs for oneself and others. In the context of intergroup relationships, these factors can enable one to picture alternative scenarios involving oneself and others—regardless of whether such interactions are real or imagined. However, despite extensive study of the perspective-taking downstream consequences, the process by which people take another's perspective and the factors which effect this process are not clearly understood [16,24,25].

Following Hoffman [26] and others, imagining oneself in the other's shoes requires a higher level of perceptual and cognitive effort. For example, comprehending the perspectives of others [27]; constructing mental representations of others to infer their thoughts and emotions [28]; evaluating concerns for the welfare of another individual [29]; or shifting between one's own perspective and that of others [30]. Moreover, prior research indicated that inferring other people's beliefs during perspective-taking requires explicit motivation rather than being automatically inferred [31]. Coincided with the above, perspective taking expected to be effortful compared with focusing on self-perspective [24].

In the context of intergroup relationships, several processes can enable one to picture alternative scenarios involving oneself and others—regardless of whether such interactions are real or imagined [32]. Recently, a three-level model that reflects the interaction between the situation and metacognitive levels during appraisal of alternative states of affairs and their effect on cognitive and emotional subjective experience has been proposed [33]. In the current study we suggest applying this model to assess perspective-taking components when imagining different perspectives of oneself and others. Situational factors may be essential to allowing one to put oneself in the shoes of others [34]. Two unique components consist of the situation level: relevancy, which relates to the extent of the scenario's significance to the individual, and plausibility, which addresses the likelihood of the individual encountering such a situation. Relevancy appraisal has been suggested as one of the basic aspects of emotional processing [35]. Indeed, ingroup bias has been found to be more evident with regard to attributes that are deemed more relevant to the ingroup members [e.g., 36, 37]. The subjective evaluation of plausibility relies on the degree to which a given state of affairs coincides with an individual's previous knowledge or experience [38,39]. This perspective suggests a potential relationship between plausibility and relevance, which may also influence the perspective-taking process.

Imaginability implies an individual's awareness and emotional reaction to the process of imagination within the framework of an imagined scenario involving oneself and other contingent events. From this viewpoint, imaginability can be categorized within the domain of metacognitive experience, which is defined as a conscious awareness pertaining to, among other aspects, the cognitive process itself [40]. Prior research suggests the capacity to imagine particular scenarios may influence plausibility evaluations [41,42]. In the realm of perspective-taking, it not only influences individual cognitive effort but also affects one's overall attitude towards different perspectives. Vividness evaluation has been proposed as a measure of the cognitive effort required in the development, manipulation, and retrieval of imaginary stimuli [43,44]. The ability to visualize outgroup and ingroup scenarios presents unique cognitive obstacles, with the degree of vividness providing as a reflection of an individual's perspective-taking cognitive processes.

Subjective experience consists of the third level of the model, which is comprised by cognitive and emotional components [33]. In the context of perspective-taking the individual requires allocated cognitive effort not just when putting on 'other shoes' but also when 'getting out'. This stage can be described as thought suppression— a range of mental control techniques to suppress spontaneous and voluntary thought, including self and other states of affairs [45]. Likewise, emotional experience has been hypothesized to be linked with the capacity to generate and manipulate an imagined perspective of others [46]. This association is also reflected in intergroup bias [47]. Therefore, categorizing others as outgroup members may reduce the relevancy of their perspective and the capacity to imagine it.

## 1.4. The current study

In the present study, we test the perspective-taking capacity by operationally identifying and distinguishing the various emotional and cognitive components that underpin imagination of ingroup and outgroup dependent states of affairs. These factors can be organized in three levels [33]. Specifically, the situational level consists of *relevancy* (*relevance to ingroup, relevance to outgroup, relevance to one's concern*) and *plausibility* factors, the cognitive level consists of *vividness* factor, and the subjective experience level consists of *emotional intensity* and *thought suppression* factors. We focus on ingroup and outgroup scenarios in the context of minorities and majorities. Recent theoretical frameworks claim that minority and majority group members experience intergroup contact differently, have different motivations and goals in

intergroup relationships, and highlight the uniqueness of their group identity in a distinct manner and degree [48]. While minority members may share some aspects of the majority identity, the opposite is rarely true for majority members [5,49]. Moreover, minority members frequently find themselves within the public sphere established by and for the majority—which may give them more opportunities than their majority counterparts to construct mental representations of outgroup-related states of affairs. Thus, minority members may require less cognitive effort than their majority counterparts when imagining majority-dominant scenarios—resulting in more vivid mental representations. Accordingly, it has been suggested that a multiple identity produces a greater willingness to engage in contact with an outgroup and may even be effective in enhancing attitudes toward the outgroup [50–52].

In the current study, we focus on the Israeli context, where the Arab resident population in Israel is the biggest minority, at approximately 20% of the total population [53]. The demographic profile inside Israel's boundaries stabilized only after the 1948 war and the 1949 ceasefire agreements with the Arab states. Although Arabs became citizens after May 1948 and are legally incorporated into the Israeli state, they are excluded from the Jewish national belonging community due to Israel's self-identification as a Jewish state and home of the Jewish people. This state of affairs can make it extremely challenging for majority and minority group members alike to imagine themselves in each other's position [54]. Furthermore, the national and ethnic identification in this context is a topic of controversy and has broad political ramifications. Specifically, studies on the Arab minority in Israel that dealt with the issue of identity were unable to reach an agreement and proposed multiple names as well as sub-groups divisions for this population (e.g., Arabs–Palestinians, Bedouin, Druses) [55]. Despite the foregoing, one of the distinctive characteristics of the whole Arab population of the State of Israel is their mother tongue, Arabic, which is recognized as the official language for the minority in Israel [56]. According to the Israeli Central Bureau of Statistics (CBS), 82% of Jews use Hebrew as their primary language at home, whereas 96% of Arabs speak Arabic [57]. As a result, we employ native language as our sample criteria, that is Hebrew native language for majority group (Jews) and Arabic native language for minority group (Arabs). In two experiments, participants were given short scripts that described four kinds of themes (see Table 1 for examples): *Neutral*; *Generic Negative*; *Identity-dependent* (Ingroup or Outgroup); or *Equally-applicable Identity-related*. It has been suggested that, when collective identity is prominent (e.g., specific mention of group membership), the distinction between ingroup and outgroup members is salient—resulting, in part, in intergroup bias [5]. Thus, in the first experiment, we expected the difference between majority and minority participants to be evident mainly in the *Identity-dependent* theme condition (see Table 1, rows 3–4). Accordingly, we hypothesized that when majority group participants are asked to imagine themselves in the shoes of the minority, they would rate the outgroup's alternative scenarios as less relevant to themselves. This would be evident in less vivid imagery and a less subjective emotional experience, as well as the effort required to suppress these imageries compared with their minority counterparts. However, the social categorization of oneself and others into ingroups and outgroups is defined as context-dependent [58]. Accordingly, we asked if the imagined situation's reliance on an exclusively specific identity is essential for the formation of a difference in perspective-taking. To answer that question, we conducted a second experiment in which participants were asked to

**Table 1. The study pool provided examples of translated scripts.**

| No. | Theme | Script |
|-----|-------|--------|
| 1 | Neutral | *I am taking five steps forward, before turning left.* |
| 2 | Generically negative | *Mom is telling me that she has cancer that is spreading throughout her entire body.* |
| 3 | Minority Identity-Dependent | *I am an Arab woman at an Israeli checkpoint, where an Israeli soldier is shouting and brandishing a rifle at me.* |
| 4 | Majority Identity-Dependent | *I'm a Jewish woman walking down the street, when a knife-wielding Arab runs toward me.* |
| 5 | Equally applicable outgroup identity-related | *When I'm abroad, people notice that I'm Arab/Jewish and refuse to sit next to me on the bus.* |

imagine themselves as outgroup members in scenarios that were equally applicable to members of either group (i.e., *Equally-applicable* themes, Table 1, raw 5).

## 1.5. Open practices and transparency

For each study, we report how we determined our sample size, all data exclusions (if any), all manipulations, and all measures in the study. Neither of the studies reported in this article were preregistered. All data and code used for stimulus presentation and analysis are publicly available on the OSF Repository at: https://osf.io/ufxzn

## 2. Method Experiment 1

### 2.1. Participants

Sixty-nine individuals, recruited from University of Haifa, took part in Experiment 1 between October 1th 2021 until November 28th 2021. Eleven individuals were excluded from analysis due to technical faults. The remaining participants (48 females, 10 males; 29 Jews, 29 Arabs; M = 25.9, SD = 6.41, age range = 19–40 years) were all native Hebrew or Arabic speakers. The inclusion criteria included having no history of diagnosed psychological disorders. The study was approved by Haifa university's Behavioral Ethics Committee (Ethics Code #184/21) on April 20th, 2021, and the participants signed an informed consent form before starting the experiment in accordance with the World Medical Association Declaration of Helsinki. They were also given a nominal fee for taking part. The data-stopping rule and sample size were determined prior to data collection. Given that the responses were on an ordinal scale, and we employed non-parametric analysis (e.g., Wilcoxon test), we computed a sample size for the two-sided Wilcoxon test when applied to two samples. The analysis was performed with a statistical power of 80% to detect a significant difference (α = 0.05) and resulted in the required 870 observations for each group to detect a minimum effect size of r = 0.122 or greater, that is 60 participants in total. Note, due to the possibility of violating the general assumptions of Cohen's formula, Rosenthal [59] proposed $r = Z/\sqrt{N}$ for analyzing Wilcoxon Signed-Rank test effect size given N = number of observations.

### 2.2. Stimuli

The experimental session consisted of 120 trials—each presenting a different 9-word script (for examples, see Table 1, rows 1.1–4.3), and seven rating tasks. Arabic-speaking participants were given scripts in Arabic, and Jewish participants were given Hebrew ones. Given that Hebrew and Arabic verbs conjugate for gender (male or female), we matched the presented scripts to each participant's gender, as stated at the start of the task. These 120 scripts were then divided into three groups, each representing a distinct theme. Thirty scripts depicted neutral scenarios, and thirty depicted generically negative ones, from a corpus validated in a previous normative research study [41]. The third group of sixty scripts centered on negative identity-dependent themes. These scripts were then examined in an online pilot study. Thirty scripts depicted potentially detrimental minority-identity-dependent scenarios, and thirty depicted potentially detrimental majority-identity-dependent ones. Specifically, in the Israeli context, the themes of these scripts were mainly about the consequences of being either Jewish or Arab.

### 2.3. Design and procedure

To encourage participants to imagine the scenarios, we employed a validated script-driven imagery task in which participants were instructed to imagine themselves in multiple. Of the range of techniques used to stimulate participants' imaginations, the script-driven imagery procedure has proven to be the most reliable and widely adopted. Specifically, script-driven imagery can effectively induce the intended mental imagery and in sparking related physiological responses akin to those encountered in genuine emotional contexts [e.g., 60–62]. Moreover, the script-driven imagery paradigm offers control over the stimuli presented to all participants, ensuring a standardized experimental framework [63]. At the

onset of each trial, participants were instructed to read the scripts and imagine themselves as active participants in the situations described therein. During the imagination stage, they could close their eyes. No time limit was imposed.

In the next stage, participants were asked to rate their experience in terms of seven aspects: *Emotional Intensity* (i.e., the intensity of their subjective emotional experience during the imagination task); *Plausibility* (the likelihood of finding themselves in the situation described in the script); *Vividness* (the ability to produce very clear and detailed mental images of situations); *Relevance to My Concerns* (the relevance of the described situation to the participant's concerns); *thought suppression* (the effort needed to disengage from the situation and the emotions it evoked, in order to move on to the next script); *Relevance to My Ingroup* (Jewish students for Jews and Arabs students for Arabs); *Relevance to My Outgroup* (i.e., Jewish students for Arabs, and Arab students for Jews). For the *Emotional Intensity* rating task, the measurement scale ranged from *1 (No emotion at all)* to *9 (Extremely emotional)*. For the *Plausibility* rating task, the measurement scale ranged from *1 (Utterly implausible)* to *9 (Extremely plausible)*. For the Vividness rating task, the measurement scale ranged from *1 (Not at all vivid)* to *9 (Extremely vivid)*. For the *Thought suppression* rating task, the scale ranged from *1 (No effort is required to suppress)* and *9 (High effort is required to suppress)*. For the *Relevance to Ingroup/Outgroup/My Concerns* rating tasks, the scale ranged from *1 (Completely irrelevant)* to *9 (Extremely relevant)* (for example timeline see Fig 1).

The experiment was carried out in the lab; instructions were simultaneously displayed on a computer screen and delivered verbally in Arabic for Arabs and in Hebrew for Jews. For each participant, the sequence of trials was randomly selected by the computer. Participants were told that they were taking part in an experiment that gauged one's ability to imagine a variety of situations, and their effect on reading. Presentation of stimuli and registration of responses were controlled by the *PsychoPy* v.2022.1.4 application, hosted on the Pavlovia platform [64]

## 2.4. Reliability of the Task

To confirm the reliability of the task, we conducted an internal consistency test on the corresponding discrete-integer-scored Likert-type items. First, we determined Guttman λ3 measure of reliability [65] and Revelle's β (i.e., the worst split-half reliability) [66]. To accomplish this, we employed the splitHalf function of the psych package (Version 2.3.6) [67]. On the strong presumption that the one-factor model is accurate for each of the rating factors in each theme condition (i.e., our item set is unidimensional), we estimated the factor-model parameters and calculated the composite reliability. Specifically, we calculated the coefficient ω for categorical items [68]. To this end, we used the *cfa* functions of the lavaan package (Version 06–16) [69] and function *reliability* of the semTools package (Version 0.5–6) [70]. In addition, we calculated Flanagan-Rulon r-equivalent reliability [71] with the random splitting method and 1000 replications. To this end, we used the *by_split* and *split_coefs* functions of the splithalfr package (Version 2.2.2) [72]. ω hierarchical measurements that have been suggested as the most comprehensible values for categorical data [73] were greater than 0.88 for all reliability estimations. A full matrix of all reliability measurements, pertinent R code, and raw data for all participants are available on the OSF Repository as Supplementary Materials.

## 2.5. Data analysis approach

The dependent variables were the ratings for *Emotional Intensity*, *Thought suppression*, *Vividness*, *Relevance to Ingroup members*, *Relevance to Outgroup members*, *Relevance to My Concerns*, *Plausibility*, and the ordinal variable of the values 1–9. The independent variables of interest were Group-Type condition (minority/majority) and Script Theme (Neutral, Generically Negative, Ingroup Identity-Dependent, Outgroup Identity-Dependent).

To determine whether there were significant differences between groups in their rating patterns of different factors within different scripts' theme, we performed the Aligned Rank Transform (ART). The ART is a nonparametric ranking procedure for ordinal and non-normal distribution data that allows conduct on the aligned data omnibus tests. First, the ordinal

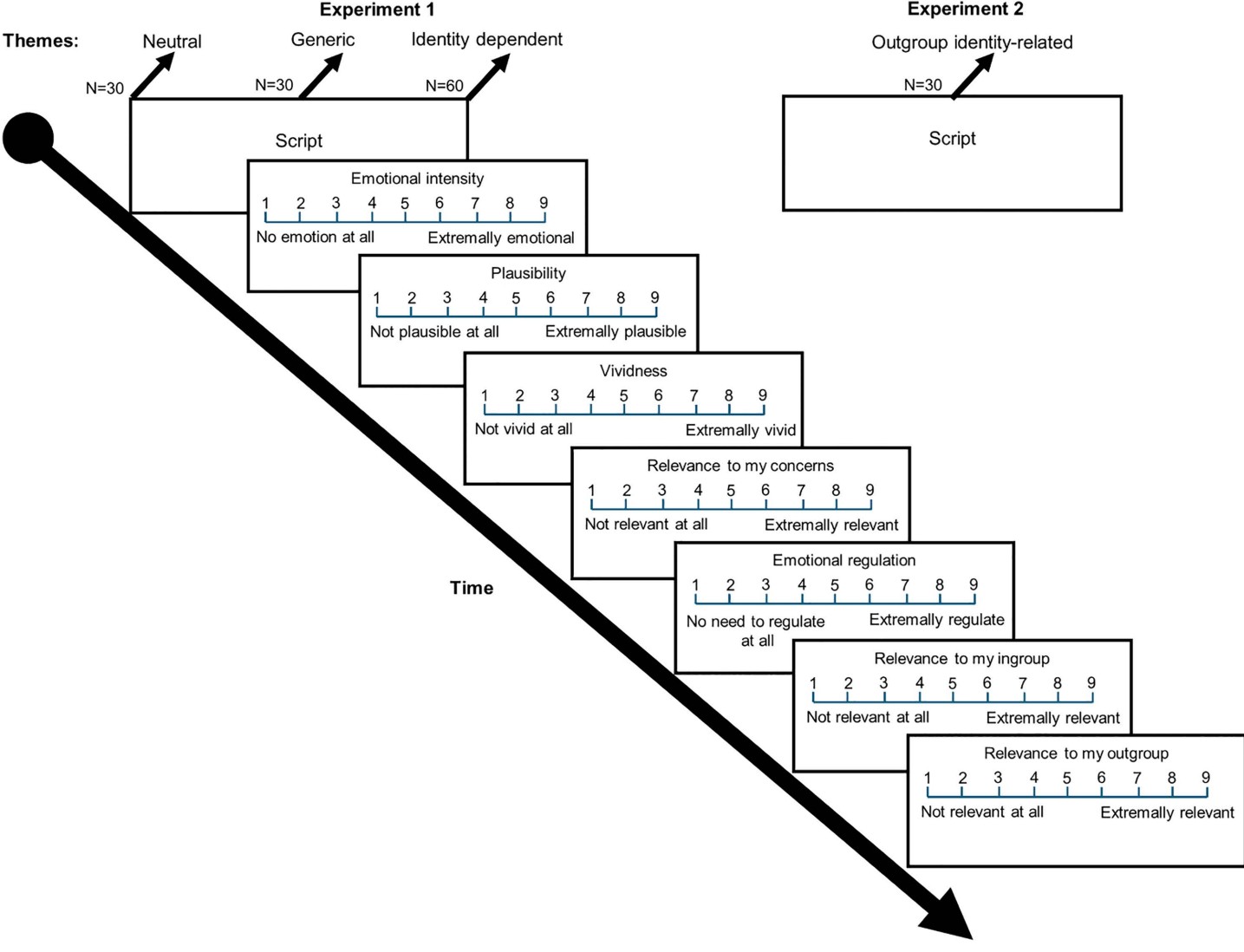

**Fig 1. Example timelines from Experiments 1 and 2.**

responses were "aligned" with respect to the specific expected effect before the mid-ranks were applied [74]. Then we use the aligned ranks to perform a linear mixed model analysis. For this aim, we used the _ARTool_ package (Version 2.3.2) in the R [75]. For better understanding patterns of rating, we performed a Bayesian ordinal regression. This approach is highly recommended as a replacement to traditional approaches that analyze ordinal data incorrectly as either nominal or interval [76]. Furthermore, choosing the Bayesian framework can provide more information about the participants' rating patterns and enables a more natural quantification of uncertainty than with the frequentist approach [77]. For this aim we used the _brms_ package (Version 2.21) in the R programming environment [78]. To improve model convergence, we used _Markov chain Monte Carlo_ (MCMC) sampling—with 4 chains of 4000 iterations, a warmup of 2000, and a weakly informative normal distribution prior to all regression coefficients [79]. Because we couldn't rule out the possibility that our predictor (group type) varies in its effects on different response categories (1–9), we fitted an _adjacent-category-logit_ model to the ordinal response data. With this approach, we could model the predictor (group type) as having category-specific effects

by estimating not one but *k* coefficients for it. We modeled both participant and item parameters through partial pooling for a more robust estimation, and to reduce the effect of extreme patterns and noise in the data [80]. For correlations between dependent variables we conducted Bayesian multivariate ordinal regression model and fitted a *cumulative-logit* model. For this aim, we employed the *brms* package (Version 2.21) [78] and the *R marginaleffects* package (Version 0.20.1) [81].

## 3. Results Experiment 1

### 3.1. Imagining identity-dependent scenarios—majority and minority

To test our hypothesis that minority participants may demonstrate a greater capacity to imagine themselves in the position of the other group compared to majority participants, we carried out a linear mixed model analysis of variance based on the Aligned Rank Transform. The results of all seven factor ratings in both identity-dependent themes (i.e., outgroup identity-dependent, ingroup-identity dependent) are presented in Table 2. For all factors, the main effect of script theme was found to be significant, and the effect sizes ranged between small ($\eta_p^2$=0.03) for *Thought suppression* and large ($\eta_p^2$=0.195, $\eta_p^2$=0.187) for *Relevance to ingroup* and *Relevance to outgroup*, respectively. The main effect for group was found significant only for *Plausibility* and *Relevance to my concerns.* However, the interaction effect between script theme and group was found to be significant for all factors except *Relevance to outgroup,* indicating that the differences between groups and their extent depend on script theme (i.e., outgroup identity-dependent, ingroup-identity dependent).

We utilized Bayesian ordinal regressions to look more closely at the differences in how the groups evaluated the various components within the different themes (Fig 2). A full table with all Bayesian ordinal regression results, including all regression coefficients and marginal effects for all factors, can be found in OSF Repository as Supplementary Materials.

**Table 2. Analysis of variance of aligned rank transformed rating data by linear mixed-effects model.**

| Factor | Effect | F | Df | Df.res | P-Valjue.adj | $\eta_p^2$ |
|---|---|---|---|---|---|---|
| Vividness | Group | 1.12 | 1 | 56 | 0.295 | 0.019 |
| Vividness | Theme | 180.46 | 1 | 3420 | <.000 *** | 0.050 |
| Vividness | Group:Theme | 30.72 | 1 | 3420 | <.000 *** | 0.009 |
| Intensity | Group | 0.29 | 1 | 56 | 0.591 | 0.005 |
| Intensity | Theme | 133.48 | 1 | 3420 | <.000 *** | 0.038 |
| Intensity | Group:Theme | 5.10 | 1 | 3420 | 0.024 * | 0.001 |
| Thought suppression | Group | 1.30 | 1 | 56 | 0.252 | 0.023 |
| Thought suppression | Theme | 107.07 | 1 | 3420 | <.000 *** | 0.030 |
| Thought suppression | Group:Theme | 35.70 | 1 | 3420 | <.000 *** | 0.010 |
| Plausibility | Group | 8.68 | 1 | 56 | 0.004 ** | 0.134 |
| Plausibility | Theme | 438.36 | 1 | 3420 | <.000 *** | 0.113 |
| Plausibility | Group:Theme | 4.00 | 1 | 3420 | 0.045 * | 0.001 |
| Relevance to my concerns | Group | 4.88 | 1 | 56 | 0.031 * | 0.080 |
| Relevance to my concerns | Theme | 214.32 | 1 | 3420 | <.000 *** | 0.058 |
| Relevance to my concerns | Group:Theme | 23.69 | 1 | 3420 | <.000 *** | 0.007 |
| Relevance to ingroup | Group | 3.25 | 1 | 56 | 0.077 | 0.054 |
| Relevance to ingroup | Theme | 829.30 | 1 | 3420 | <.000 *** | 0.195 |
| Relevance to ingroup | Group:Theme | 37.53 | 1 | 3420 | <.000 *** | 0.010 |
| Relevance to outgroup | Group | 2.50 | 1 | 56 | 0.116 | 0.043 |
| Relevance to outgroup | Theme | 787.84 | 1 | 3420 | <.000 *** | 0.187 |
| Relevance to outgroup | Group:Theme | 1.70 | 1 | 3420 | 0.18 | 0.000 |

Note: Type III Wald F tests with Kenward-Roger df.

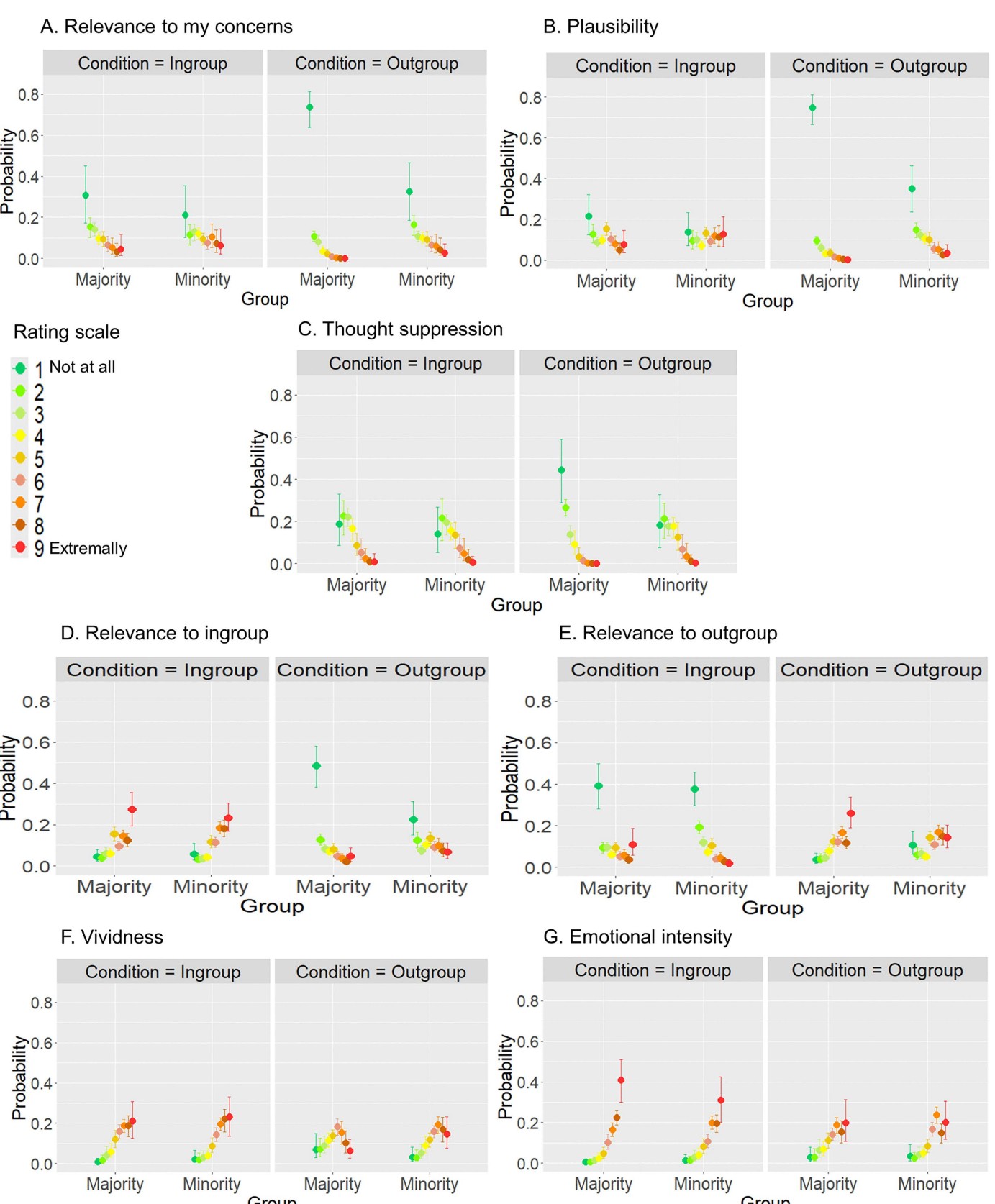

**Fig 2. Marginal effects of group on Relevance to my concerns (A), Plausibility (B), Thought suppression (C), Relevance to ingroup (D), Relevance to outgroup (E), Vividness (F), Emotional intensity (G) from model fit.**

For *Plausibility*, *Relevance to my concerns*, *Thought suppression*, and *Relevance to ingroup* notable variations were observed in the outgroup identity-dependent theme, particularly in the higher probability of selecting category-1 ("not at all") in the majority group compared to the minority. On the other hand, the difference between minority and majority ratings in the ingroup identity-dependent theme was negligible. Specifically, for *Relevance to my concerns*, the majority and minority groups differed by their ratings of ingroup identity-dependent scripts in category-7 alone (3.9%, CI = [0.1%, 7.4%]) as shown in Fig 2a. However, when it comes to scripts that rely on outgroup identification, the majority is notably more likely to select category-1 (33.3%, CI = [19.8%, 45.2%]), with modest variations observed in all other categories. The rating pattern for the *Plausibility* factor exhibited the very same pattern (Fig 2b). Namely, there was a single distinction (5.6%, CI = [1.5%, 9.9%]) in the probability of selecting category-8 for ingroup identity-dependent theme with a higher probability for the majority to select category-1 (33.8%, CI = [21.6%, 44.2%]) and modest variations in all other categories for outgroup identity-dependent theme.

No difference was observed in the ingroup identity-dependent theme for *Thought suppression* (Fig 2c). In contrast, there were slight variations in the probability of selecting categories 5, 6, and 7, whereas there was a higher probability for the majority to select category-1 (17.4%, CI = [3.4%, 30.7%]) in the outgroup identity-dependent theme. For *Relevance to ingroup* (Fig 2d), there were minor variations noted in categories 3, 7, and 8. However, the overall trend in ratings was quite similar for the ingroup identity-dependent theme across groups. Similar to the patterns mentioned above, there was a higher probability for the majority to select category-1 in the outgroup identity-dependent theme (21.2%, CI = [10.3%, 31.1%]). Although there were significant interaction effects for *Vividness* (Fig 2f) *and Emotional intensity* (Fig 2g) the small effect size predicted all creditable intervals (CI [2.50%, 97.50%]) of the differences between majority and minority groups included 0. *Relevance to outgroup* (Fig 2e) was the only factor in which the interaction and the main effect for group were found to be negligible. That is, both groups showed a higher probability of selecting category-1 (not relevant at all) when they evaluated the relevance outgroup members attribute to their ingroup theme (27%, CI = [24.6%, 29.6%]) compared to the rating of others' identity-dependent scripts. Based on the foregoing, the current findings imply that majority group members evaluate that it is less likely that they would find themselves in circumstances strongly associated with outgroup identification, minimize their relevancy to themselves, and allocate less effort to suppress thinking about them, than minority members. However, both minority and majority members tend to similarly rate their *emotional experience* and their imagery *vividness* higher when triggered by events highly connected with ingroup identification compared to events highly connected with outgroup identification. The maximum R-hat of any model's parameter was 1.0—indicating good MCMC convergence. Both bulk-ESS and tail-ESS of any parameter were higher than 400 (Bulk_ESS > 949, tail-ESS > 1808). Given that we used four Markov chains, this indicated that estimates of respective posterior quantiles were reliable.

The posterior mean estimates of the probability of responses in each rating category are shown for each of the two groups (Minority – Arabs, Majority – Jews). Error bars indicate 95% credible intervals. The factor categories range from 1 (Not at all) to 9 (Extremely).

## 3.2. Imagining neutral and generically negative scenarios

Our hypothesis posits that, in the absence of explicit identity framing, both majority and minority participants will evaluate the relevance of the situation to themselves (ingroup) and to others (outgroup) in a highly comparable manner. We employed a Bayesian ordinal regression model (adjacent category logit) to assess the correlation between rating patterns for *Relevance to outgroup* and *Relevance to ingroup* for each group (minority, majority) separately with a fixed effect of

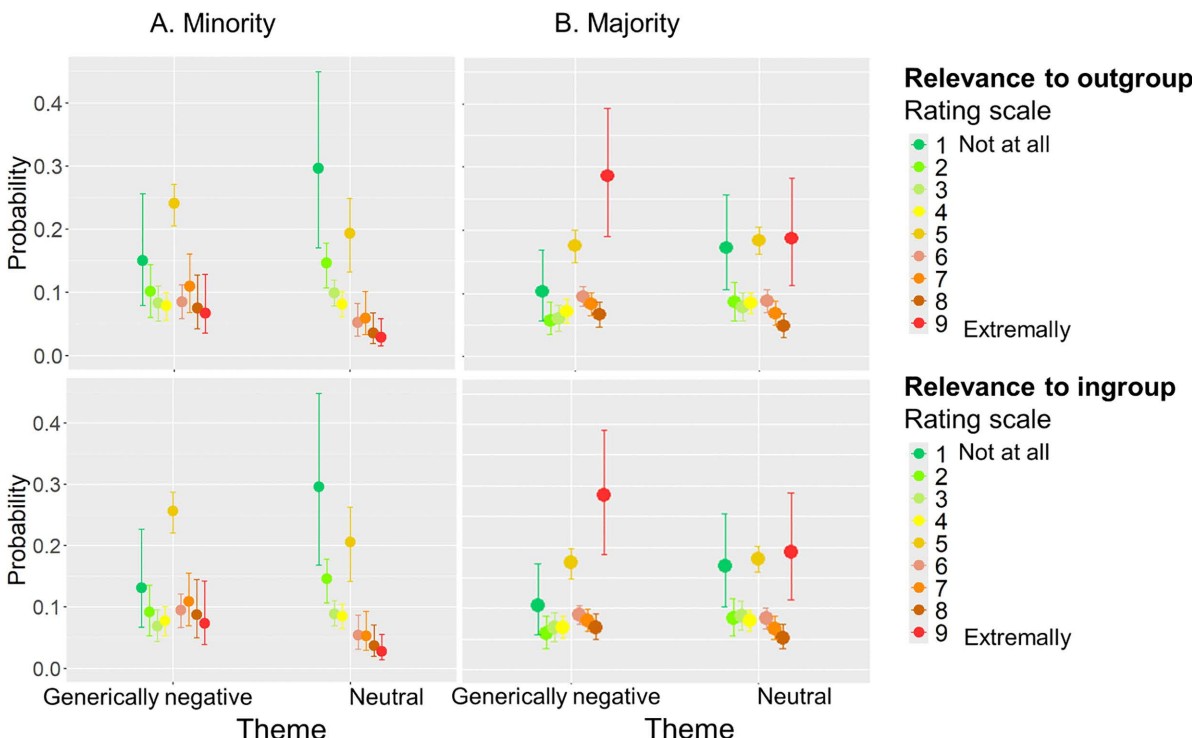

Theme (neutral, generically negative). Importantly, we allowed the model to estimate the correlation between the participants' random intercepts across the two ratings. The results showed a very strong positive correlation between participants' intercepts on the two variables for both minority and majority: cor(ingroup, outgroup) = 1.00, 95% CI [0.99, 1.00]. Moreover, for all rating categories, the 95% credible intervals encompassed zero (i.e., CI low < 0 < CI high), suggesting that no reliable differences emerged at the 95% credibility threshold. Fig 3 indicates that for both neutral and generically negative themes and for both minority and majority groups the rating patterns for *Relevance to outgroup* and *Relevance to ingroup* war almost identical. A full table with all Bayesian ordinal regression results, including all regression coefficients and marginal effects for all rating categories, can be found in OSF Repository as Supplementary Materials.

In addition to the general relevancy factors (i.e., *Relevance to outgroup* and *Relevance to ingroup)*, our study design included specific relevancy component, that is *Relevance to my concerns. Plausibility* factor is also found to be closely related to relevancy when participants evaluated emotional events [33]. Importantly, estimations of *Relevance to my concerns* and *Plausibility* effect individual experience for emotional, but not neutral events [82]. Therefore, we utilized linear mixed model analysis of variance, based on the Aligned Rank Transform to examine whether minority and majority groups evaluated generically negative events differently in terms of their *Relevance to my concerns* and *Plausibility*. In both factors the difference between majority and minority groups was significant. That is, $F = 4.67$ ($p = 0.035$, $\eta_p^2 = 0.077$) and $F = 6.08$ ($p = 0.017$, $\eta_p^2 = 0.098$) for *Relevance to my concerns* and *Plausibility*, respectively. We utilized Bayesian ordinal regressions to look more closely at the differences in how the groups evaluated the various components within the different themes (Fig 4). An examination of Fig 4a and Fig 4b suggest one notable distinction among the nine rating categories, namely the probability of selecting category-1 in *Plausibility* and *Relevance to my Concerns* ratings. More precisely, the majority group had a 13.2%, CI = [5.4%, 20.6%] higher probability of evaluating generically negative events

**Fig 3. Marginal effects of group on Relevance to outgroup (upper raw) and Relevance to ingroup (lower raw) from model fit.**

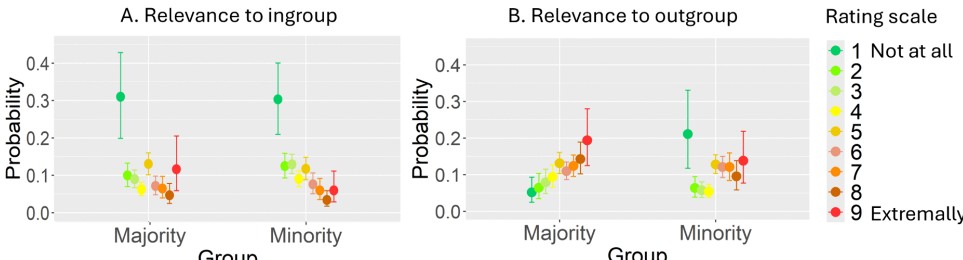

**Fig 4. Marginal effects of group on Relevance to my concerns (A) and Plausibility (B) from model fit.**

as completely irrelevant, and a 10.3%, CI = [3.0%, 17.4%] higher probability of considering them as not feasible compared to the minority group. The maximum R-hat of any model's parameter was 1.0—indicating good MCMC convergence. Both bulk-ESS and tail-ESS of any parameter were higher than 400 (Bulk_ESS > 1721, tail-ESS > 3161). Given that we used four Markov chains, this indicated that estimates of respective posterior quantiles were reliable. A full table with all Bayesian ordinal regression results, including all regression coefficients, can be found in OSF Repository as Supplementary Materials.

The posterior mean estimates of the probability of responses in each rating category are shown for each of the two groups; Minority (A) and Majority (B). Error bars indicate 95% credible intervals. The factor categories range from 1 (Not at all) to 9 (Extremely).

The posterior mean estimates of the probability of responses in each rating category are shown for each of the two groups (Minority – Arabs, Majority – Jews). Error bars indicate 95% credible intervals. The factor categories range from 1 (Not at all) to 9 (Extremely).

### 3.3. Interim discussion: Experiment 1

The first experimental matrix design comprised participant groups categorized by two levels (minority and majority) as the between-subjects factor, and three sets of scripts characterized by three levels (neutral, generically negative, and identity-dependent themes) as the within-subjects factor. We examined each group of the script individually. An additional two-level identification factor (i.e., ingroup, outgroup) has been incorporated for the identity-dependent theme analysis. Prominent interaction effects ware identified among four dependent factors: *Relevance to my concerns*, *Relevance to ingroup*, *Plausibility*, and *Thought suppression*. Namely, the majority group exhibited had a stronger tendency to select category-1 than the minority group, however this was evident solely in the outgroup-dependent themes. In simple terms, members of the majority group were more inclined to dismiss the likelihood of encountering outgroup-dependent scenarios and to diminish both the general and specific relevance of such situations to their concerns, in contrast to minority group members. Furthermore, members of the majority group were more inclined to regard outgroup-dependent situations as less challenging to suppress than members of the minority group. Nonetheless, these disparities vanished when individuals from both groups were prompted to envision themselves in events dependent on ingroup identity.

Interestingly, these patterns were not observed in the *Emotional experience* and *Vividness* rating patterns. That is, both the majority and minority reported on a distinct pattern of evaluations for emotional and vivid experiences generated by ingroup and outgroup identity-dependent events. These results suggest that both groups are more likely to select higher category rank (i.e., stronger emotions and more vividness) in the ingroup compared to the outgroup identity-dependent condition.

To account for the potential of an overall variation in situation levels components (i.e., relevancy and plausibility) across groups or intergroup biases when rating relevance to me (ingroup) or to others (outgroup), regardless of the script's

content, we add two control conditions to the item's corpus. That is, scripts that described neutral physical actions ("I am taking five steps forward, before turning left") and scripts that depicted generically negative events with no reliance on minority-majority identification ("Mom is telling me that she has cancer that is spreading throughout her entire body"). Our results revealed within each group (majority and minority), the pattern of *Relevance to outgroup* and *Relevance to ingroup* ratings for control items was nearly comparable. Specifically, without a clear framing of the group identification contained within the event characterization, participants did not make distinct judgments for the ingroup or outgroup.

When participants were asked to evaluate the plausibility of experiencing generically negative events such as illnesses or love disappointments, the majority group exhibited a stronger tendency to dismiss this plausibility compared to the minority group (i.e., choosing category-1, "not feasible at all"). These findings may indicate that the minority group may be more likely to anticipate the occurrence of general negative events in their lives. However, no statistical difference in *Plausibility* ratings of negative ingroup-dependent identity events across groups was found. A similar response pattern was found for *Relevance to my concerns.* Namely, the majority group's probability to reduce the relevancy of the generically negative and outgroup identity-dependent negative events to its concerns was higher compared to the minority group. No such differentiation was found for the ingroup-dependent negative theme.

Previous research indicated that explicit focus on self or other identification may play in intergroup bias [e.g., 83, 84]. However, in Exp. 1 the control items (i.e., neutral and generically negative) differed substantially in their content from the identity-dependent items. Specifically, the neutral scripts depicted basic physical activity, whereas the negative scripts displayed undesirable occurrences that impact Jews and Arabs alike (e.g., health worries, family difficulties). In contrast, the events described in the identity-dependent items were fundamentally outcomes of the subject's identity. As a result, it is unclear if prominent framing of an identity is sufficient to produce biases in judgments of others, or what role the event's distinctiveness and exclusive reliance on the subject's identity play in producing the effect. To answer that question, we conducted a second experiment in which participants were asked to imagine themselves as outgroup members in identity-dependent scenarios that were equally applicable to members of either group (i.e., Equally-applicable themes).

## 4. Method experiment 2

### 4.1. Participants

In Experiment 2, a total of sixty individuals (41 females, 19 males; 30 Jews, 30 Arabs; M = 26.2, SD = 5.41, range = 19–65 years) who were recruited online, took part in an online experimental procedure between December 15th 2021 to January 3th 2022. The study was approved by Haifa university's Behavioral Ethics Committee, and the participants signed an informed consent form before starting the experiment. They were also given a nominal fee for taking part.

### 4.2. Stimuli, design, procedure and data analysis approach

The experimental session consisted of 30 trials, in which different 9-word negative scripts were presented (for example, see Table 1, row 5). We specifically developed two equally-applicable script versions for this study, with the only difference being whether the protagonist is Jewish or Arab. In each script, the protagonist's Jewish or Arab identity was stated explicitly—i.e., "I am Jewish," or "I am an Arab." Each of the groups was presented with outgroup-related themes—namely, Jewish participants were asked to imagine themselves as Arabs, and vice versa. Unlike the identity-dependent themes of Experiment 1, Experiment 2's scripts were equally applicable to Jews or Arabs. All scripts in both experiments were written in the first-person singular form, allowing participants to read the script more closely to their internal speech and thereby imagine themselves as protagonists in these alternative scenarios.

The design was identical to Experiment 1. However, the debriefing of participants for Experiment 1 indicated that the Plausibility rating task in the Outgroup theme condition (i.e., Jewish themes for Arabs and Arab themes for Jews) may be

interpreted in several ways. We therefore added an explicit clarification to Experiment 2, in the form of the question, "How likely is it that you would find yourself in the situation described in the scripts if you were Jewish [for Arab participants]/ Arab [for Jewish participants]?". Because the experiment took place online, participants only received instructions on a computer screen. Participants spent between 25 and 35 minutes in a session. The data analysis approach was identical to Experiment 1. The independent variable of interest was only the Group-Type condition (minority/majority).

### 4.3. Results

The linear mixed model analysis of variance, based on the Aligned Rank Transform, indicated only one marginally statistically significant effect of group on *Relevance to outgroup* factor (Table 3). Additional Bayesian ordinal regression for *Relevance to outgroup* and *Relevance to ingroup* has been performed. A full table with all Bayesian ordinal regression results, including all regression coefficients, can be found in OSF Repository as Supplementary Materials. Inspection Fig 5a indicated that, while there was a little variance in categories 3, 4, and 9 (e.g., CI [2.50%, 97.50%] did not include 0), the general trend in ratings for *Relevance to ingroup* was relatively comparable across groups. That is, a higher probability of selecting category-1 (30.5%, CI = [20.6%, 40.3%], 31.4%, CI = [20.0%, 42.5%], for the majority and minority, respectively) than all other categories. Fig 5b reveals that when participants evaluated *Relevance to Outgroup*, the minority was more likely to select category-1 in 17.8% (CI = [10.0%, 25.4%]). Conversely, the majority showed a positive correlation between the probability of selecting a category and its rank, with the probability of selecting category-9 (19.4%, CI = [12.5%, 28.0%]) being similar to the minority's probability of selecting category-1. The maximum R-hat of any model's parameter was 1.0—indicating good MCMC convergence. Both bulk-ESS and tail-ESS of any parameter were higher than 400

**Table 3. Type III Wald F tests with Kenward-Roger df.**

| Factor | F | Df | Df.res | p-Value.adj | $\eta_p^2$ |
|---|---|---|---|---|---|
| Vividness | 1.54 | 1 | 58 | 0.220 | 0.026 |
| Intensity | 0.75 | 1 | 58 | 0.390 | 0.013 |
| Thought suppression | 0.36 | 1 | 58 | 0.549 | 0.006 |
| Plausibility | 0.20 | 1 | 58 | 0.653 | 0.004 |
| Relevance to my concerns | 0.02 | 1 | 58 | 0.894 | 0.000 |
| Relevance to ingroup | 0.27 | 1 | 58 | 0.607 | 0.005 |
| Relevance to outgroup | 3.33 | 1 | 58 | 0.073 | 0.054 |

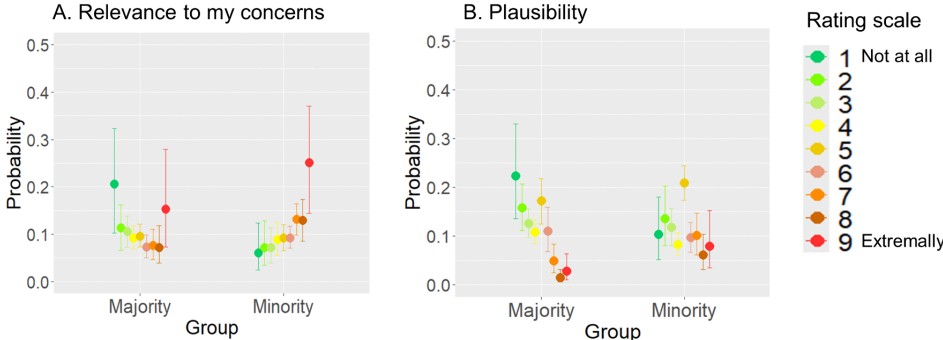

**Fig 5. Marginal effects of group on Relevance to my ingroup (A), Relevance to outgroup (B), from model fit.**

(Bulk_ESS > 2410, tail-ESS > 4121). Given that we used four Markov chains, this indicated that estimates of respective posterior quantiles were reliable.

**Analysis of variance of aligned rank transformed rating data by linear mixed-effects model**

The posterior mean estimates of the probability of responses in each rating categories are shown for each of the two groups (Minority – Arabs, Majority – Jews). Error bars indicate 95% credible intervals. The factor categories range from 1 (Not at all) to 9 (Extremely).

### 4.4. Interim discussion: Experiment 2

The results of Experiment 1 suggested that when participants were asked to imagine themselves as outgroup members (Majority as Minority and vice versa) in identity-dependent scenarios, a prominent difference was found between groups in four factors. Namely, the majority group members were more likely than the minority to evaluate outgroup-dependent situations as completely irrelevant to their concerns and to their ingroup, to disregard the plausibility that they had found themselves in an outgroup-dependent situation, and accordingly to allocate less cognitive effort to suppress thought regarding these outgroup-dependent situations than the minority.

These findings coincided with the claim that situational accessibility (i.e., relevancy of an individual's current social context) affects the salience of social identity, which in turn can lead to enhanced intergroup bias [85,86]. Therefore, we predicted that controlling the relevancy component of the alternative states of affairs that participants were asked to generate in their minds would minimize the distinction between the groups. As expected, reducing the reliance of the scenario on exclusively one identity—i.e., making the circumstances equally applicable to majority and minority group members—reduced the distinction between the groups. Specifically, only one marginal difference was found in *Relevance to outgroup* component. That is, the minority tended to evaluate the relevancy of a specific situation to themselves and others in a similar direction (i.e., not relevant). On the contrary, the majority evaluated the same scenarios as more relevant to others than to themselves, although the scripts were based on Equally-applicable themes.

## 5. Discussion

In this study, we examine how majorities (i.e., Jews) and minorities (i.e., Arabs) in Israel evaluated various aspects of their imagined experience while switching between imaginations of scenarios related to ingroup and outgroup members. We focus on several components that underpin imagination of ingroup and outgroup dependent states of affairs. Namely, *relevancy* (*relevance to ingroup, relevance to outgroup, relevance to one's concern*) and *plausibility* factors consists the situational level; *vividness* factor consists the cognitive level; *emotional intensity* and *thought suppression* factors consists the subjective experience level. Previous studies found that imagined intergroup contacts may affect levels of perspective-taking and reduce bias toward a range of outgroups [87]. Furthermore, vividness of imagery and adopting the outgroup's perspective mediated the association between stimulated alternative scenarios and subjective experience [88,89]. In two experiments, participants rated various imagined scenarios described in written scripts in terms of relevance, plausibility, and cognitive and emotional experience.

Inspired by Sherif (RGCT) [13], Tajfel (SIT) [90], and the extensive ensuing research tradition, we expected to find differences in components of imagination, primarily when participants were required to evaluate various scenarios in terms of their group identification. Coinciding with our predictions within each group (majority and minority), without a clear framing of the group identification contained within the event characterization (i.e., neutral and generically negative conditions), participants did not make distinct judgements for the ingroup or outgroup. That is, no meaningful and robust difference was found between the majority and minority participants in imagination components (i.e., emotional, meta-cognitive, and situational). Moreover, the difference between minority and majority ratings in the ingroup identity-dependent theme was negligible.

Given that groups in conflict tend to exhibit greater ingroup bias, less intergroup affinity, and greater intergroup discrimination [3,91], we expected both groups to exhibit marked differences in imagination when asked to switch perspectives between ingroup and outgroup in their imagination. Sure enough, members of both groups explicitly reported having difficulty imagining themselves in scenarios that more clearly characterized their respective outgroup. This difficulty was anticipated in light of the presence of symbolic threats (ITT) [92] that delineate differences in worldviews, values, and identity between the Israeli majority and minority. Critically, building on previous studies that found differences between majority and minority members with regard to intergroup relations [9] and the inverse relations between power and perspective taking [93–95], we predicted that majority participants would tend to view minority-identity-dependent themes as highly irrelevant to themselves, less feasible, and less trigged emotionally in contrast to ingroup-identity-dependent themes.

Our results revealed that the difference between minority and majority in the ingroup identity-dependent condition was negligible. Furthermore, the majority and minority groups did not differ in rating neutral or generically negative events as relevant to themselves or others. These findings imply that when participants were not required to deliberately take an outgroup viewpoint (imagine yourself as an outgroup member in the first person), they did not display explicit bias in outgroup judgment. However, when participants were asked to imagine themselves in a variety of scenarios based on exclusively outgroup membership, marked differences were found between the two groups. Both groups notably chose category-1 for outgroup-dependent themes when evaluating their *Plausibility*, *Relevance to my concerns*, *Thought suppression*, and *Relevance to ingroup* variables. Nevertheless, in accordance with our predictions, the majority group exhibited a far higher tendency to do so. Simply put, the majority was more likely to disregard the plausibility that they had found themselves in an outgroup-dependent situation than the minority. They also found these situations less relevant to themselves and easier to suppress.

Interestingly, these patterns of response did not replicate in Emotional intensity, and Vividness components. Specifically, we hypothesized that minority participants would have an advantage over majority participants in their capacity to generate and maintain outgroup identity-dependent motifs in their imagination. These advantages may impact their emotional response. However, we found no noticeable difference between the groups. The lack of effect in both variables can be attributed to a strong reluctance within the Israeli-Palestinian national conflict to openly acknowledge elements of perspective taking that could be perceived as emotional and cognitive empathy. In Experiment 2, we reduced the reliance of the scenarios on exclusively one identity. Thus, although these were identity-dependent states of affairs, making the circumstances equally applicable to majority and minority group members reduced the distinction between the groups. These findings suggest that the effect of prominent group identity framing on putting oneself in outgroup shoes and in turn, on intergroup bias may be moderated by situational accessibility factors. For example, an identity-based theme in which Jews/Arabs feel unwelcome in Western nations is prevalent in both Arab and Jewish popular discourse [96,97]. Despite the fact that the topic is identity-dependent, it is also accessible to Jews and Arabs. In other words, the precise scenario in which one tries to put themselves in the shoes of an outgroup member is critical to the effectiveness of this process. Despite the significant role that situational factors may play in perspective taking, as indicated by previous research, the lack of data from naturalistic studies impedes our comprehension of the specific ways in which contextual factors affect perspective taking [34]. In the current study, we take a step to close this gap. Our findings imply that stepping into the shoes of others is prone not just to the distinctive characterization of the perspective taker (e.g., minority or majority), but also to the sole dependency of the particular scenario on the identification of the other.

Comprehending the complex relations between minority and majority groups within the social hierarchy is a major challenge for social psychology researchers, considering the crucial impact these interactions have on the structure of contemporary societies. In the Israeli context, some of the major factors that characterize minority and majority relations are revealed in their extreme phase. Specifically, distinct group identities; numerous interactions between the groups in everyday life on the one hand, and strong limits on the merger of the groups on the other; significant disparities in access to resources and power in the context of violent and ongoing national conflict. We propose that, under certain conditions,

minority members are less susceptible to intergroup bias. Some of these conditions are experienced by Israel's Arab minority in real life. First, their identity can best be described as a complex repertoire of identities comprising various components (e.g., Arab, Palestinian, Israeli), in contrast to the cohesive identity of Israel's Jewish majority [98]. That is, while the Arab minority and Jewish majority share some aspects of Israeli identity, the Jewish majority not only shares no aspect of Arab identity whatsoever but also vehemently opposes it [99,100]. Moreover, the Arab minority interacts frequently with the Jewish majority, is bilingual, and is highly exposed to Jewish-Israeli culture, and thus may be more accustomed to conjuring up mental representations of Jewish scenarios in their minds than the other way around.

*Permeability* is defined as the likelihood of group members shifting from one group to another [6,101]. This is almost impossible in the Israeli Arab-Jewish context, in which the two populations cannot intermingle, for national and religious reasons. However, given that members of lower-status groups (minorities) may tend to covet membership in higher-status groups (majorities), they would presumably also find it easier to imagine themselves as majority members than vice versa. The majority and minority groups essentially differ in their social power, which pertains to their control over valued tangible and symbolic resources. This aspect is separate from a social status and is considered a pivotal component in the human hierarchy, having significant implications for individual behavior [102]. Specifically, it has been suggested that power reduces perspective-taking and concern for others [92]. Our findings may support this premise, given that national and ethnic minorities, such as those in Israel, are often positioned at the bottom of the social hierarchy, with limited access to economic, political, and cultural power reserves.

## 6. Conclusion

The rise in global migration, coupled with growing nationalist sentiment, is among the greatest challenges facing humanity in the early twenty-first century. Societies that were considered homogeneous fifty years ago have become multicultural, multiethnic, and multireligious in the past two decades. These societies, which are structured as group-based hierarchies, have established a complex asymmetrical relationship between minorities and majorities. Here, we show that during an imagination paradigm, when participants are asked to imagine themselves in the other's shoes, minority and majority differ in some of the major aspects of the process. Specifically, majority members (such as Jews in Israel) experienced outgroup-related themes less relevant and less plausible to themselves compared to minority members. Accordingly, the majority members required less cognitive effort for suppressing outgroup-related themes than vice versa. This may suggest that minority members (such as Arabs in Israel) are more capable or more motivated to temporarily adopt alternative perspectives, even in the context of intergroup conflict and significant differences in social power. Enhancing our understanding of this topic would facilitate the development of effective educational and social strategies to improve intergroup interactions.

## 7. Limitations

Our study faces several limitations that highlight areas for future research. A) Notably, stereotypes and attitudes towards outgroups are often context-specific. This is particularly salient in contexts of national conflicts, such as the Israeli-Palestinian conflict [103]. Contextual factors may also influence whether ethnic and national groups self-identify as a minority or a majority. For example, in the Israeli–Palestinian context, scholars have proposed distinguishing the national minority within Israel from the regional majority in the Middle East, a framing that may affect how the two groups perceive one another [104]. Consequently, there's a need for future studies to test the generalizability and relevance of our findings in non-Israeli populations and in varied social contexts to further validate our results. B) The inherent variability within appropriated groups might impact intergroup dynamics. For example, within the Israeli context, the social hierarchy of the four primary subgroups comprising the Arab community—Bedouin Arabs, Muslim Arabs, Druze, and Christian Arabs—also exerts an impact on their interactions with the Jewish majority [105]. Thus, in follow-up research, a more detailed examination of subgroups is necessary. C) Previous research has revealed gender differences in perspective

taking—specifically, that females may have a greater capacity to comprehend the mental state of others [e.g., 106, 107]. Given the preponderance of female participants in the current studies, larger and gender-balanced samples are required to further validate the findings reported here. D) The current study was based on self-report measurement exclusively. Thus, follow-up study with additional implicit, physiological, and brain measures is needed. E) Given our method and the multi-trials task we used, the statistical power of the study was sufficient; however, a larger sample size in the following studies is required to establish the robustness of our findings.

## Supporting information

**S1 File. Inclusivity in global research questionnaire.**
(DOCX)

## Author contributions

**Conceptualization:** Dalit Milshtein, Ahmad Serhan, Simone Shamay-Tsoory.

**Data curation:** Ahmad Serhan.

**Formal analysis:** Dalit Milshtein.

**Methodology:** Dalit Milshtein.

**Supervision:** Simone Shamay-Tsoory.

**Visualization:** Dalit Milshtein.

**Writing – original draft:** Dalit Milshtein.

**Writing – review & editing:** Simone Shamay-Tsoory.

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
