## [Decision Letter · Decision Letter 0]

24 Sep 2025

Dear Dr. Milshtein,

Thank you for submitting your manuscript to PLOS ONE. After careful consideration, we feel that it has merit but does not fully meet PLOS ONE’s publication criteria as it currently stands. Therefore, we invite you to submit a revised version of the manuscript that addresses the points raised during the review process.

Both reviewers found your manuscript strong and suitable for publication. Reviewer 1 recommended acceptance as is, while Reviewer 2 raised a few important points regarding the definition and description of the study population, specifically the term “Israeli-Arabs.” Please revise the manuscript to address these comments and resubmit for further consideration.

We look forward to receiving your revised manuscript.

Kind regards,

Gal Harpaz, Ph.D.

Academic Editor

PLOS ONE

Additional Editor Comments (if provided):

Reviewers' comments:

Reviewer's Responses to Questions

**Comments to the Author**

1. Is the manuscript technically sound, and do the data support the conclusions?

Reviewer #1: Yes

Reviewer #2: Yes

2. Has the statistical analysis been performed appropriately and rigorously?

Reviewer #1: Yes

Reviewer #2: Yes

3. Have the authors made all data underlying the findings in their manuscript fully available?

Reviewer #1: Yes

Reviewer #2: Yes

4. Is the manuscript presented in an intelligible fashion and written in standard English?

Reviewer #1: Yes

Reviewer #2: Yes

Reviewer #1: This paper meticulously investigates the relationship between an individual's membership in a majority or minority group and their capacity for intergroup empathy. The literature review is comprehensive, current, and effectively integrates various theoretical dimensions of perspective-taking and imagining the opponent's point of view.

The methodology employed a series of experimental sessions with Arab and Israeli students at Haifa University, using script-driven imagery tasks. The results indicate that both groups had trouble imagining themselves in scenarios that distinctly characterized their respective out-groups. However, a notable difference emerged between the majority and minority participants: under specific conditions, minority members demonstrated less susceptibility to intergroup bias. Conversely, majority members perceived out-group-related themes as less imaginable. The findings suggest that an individual’s social power may influence their ability to imagine the perspective of others, which in turn contributes to in-group bias.

The paper provides a detailed account of the experimental design, presenting the results and analyses vividly through both explanatory text and illustrative charts.

The following anecdotal example illustrates the central argument of this paper regarding the influence of social position on perspective-taking and cognitive bias. At a reunion of veteran Israeli news reporters, male journalists nostalgically recalled their younger years and professional experiences. Their idealized memories were starkly contrasted by a female colleague, who, in a moment of candor, shared her negative experiences with sexual harassment at the same television channel where they had all worked together. The male reporters' subsequent silence and outright denial of her account highlight their inability to mentally accommodate her negative experiences into their own positive, self-affirming narratives. This incident underscores how a socially dominant position can constrain the capacity for out-group perspective-taking, thereby reinforcing in-group bias and obstructing the recognition of opposing realities.

As previously stated, the article is well-written and scientific, and certainly worthy of publication. Having said all this in its praise, and in the hope that it will indeed be published, I would like to add a comment for consideration. The paper's point of departure is that the Jewish group is the majority and the Arab group is the minority. This is certainly true, given the fact, noted in the article, that Arab citizens constitute 20% of the total population in the State of Israel. At the same time, it is difficult to isolate civilian life in Israel from the Arab-Israeli conflict in which, across the Middle East, Israel is often the inferior side, and certainly when it comes to numbers. Some even claim that part of the conflict stems from an Israeli sense of threat. Israel's opponents argue that this feeling is not realistic, while a large portion of Israel's supporters see the threat of the Arab states surrounding it as a real threat. Therefore, in the Israeli case, it is advisable to treat majority and minority relations with caution, even if it is an Arab minority within a Jewish state; at the end of the day - in the larger geographical area there is a basis for the claim that it is the Jews who are the minority. All this requires, in my opinion, further reflection, and perhaps such a reflection could be an outcome of the article and a basis for further research.

Reviewer #2: Dear Authors,

Thank you very much for the opportunity to review your manuscript.

I found the paper highly impressive. However, I have three related comments:

1. I did not find any background on the Israeli–Palestinian conflict. Providing some context on this issue would help readers unfamiliar with it better understand the importance of the study.

2. In the manuscript, you refer to the Arab participants as “Israeli Arabs.” Did you ask the participants about their self-identification? In many publications this group is referred to as “Arabs–Palestinians who live in Israel,” so it may be worth clarifying or justifying your choice of terminology.

3. It would also be helpful to include a clear definition of how the majority of this group self-identifies, so that readers understand whom the term refers to.

I hope these comments will be helpful to you in strengthening your manuscript.

**Do you want your identity to be public for this peer review?** For information about this choice, including consent withdrawal, please see our Privacy Policy

Reviewer #1: **Yes: ** Prof. Eyal Lewin

Reviewer #2: No

---

## [Author Response · Author response to Decision Letter 1]

13 Oct 2025

Our answer to academic editor and Journal requirements:

1. We ensured that our manuscript meets PLOS ONE's style requirements.

2. We included a complete copy of PLOS’ questionnaire on inclusivity in global research in our revised manuscript as Supporting Information.

3. We included a separate caption for each figure in our manuscript and uploaded the figures as separate files.

4. We reviewed our original reference list. No changes have been done. However, we change the formant to "Vancouver” style as required. The additional references marked in the revised manuscript

Reviewer #1

1. I would like to add a comment for consideration. The paper's point of departure is that the Jewish group is the majority and the Arab group is the minority. This is certainly true, given the fact, noted in the article, that Arab citizens constitute 20% of the total population in the State of Israel. At the same time, it is difficult to isolate civilian life in Israel from the Arab-Israeli conflict in which, across the Middle East, Israel is often the inferior side, and certainly when it comes to numbers. Some even claim that part of the conflict stems from an Israeli sense of threat. Israel's opponents argue that this feeling is not realistic, while a large portion of Israel's supporters see the threat of the Arab states surrounding it as a real threat. Therefore, in the Israeli case, it is advisable to treat majority and minority relations with caution, even if it is an Arab minority within a Jewish state; at the end of the day - in the larger geographical area there is a basis for the claim that it is the Jews who are the minority. All this requires, in my opinion, further reflection, and perhaps such a reflection could be an outcome of the article and a basis for further research.

Our response:

We thank the editor for this important comment. Indeed, in the Israeli context, self-identity as minority or majority may be influenced by variables other than the number of Jews and Arabs within Israel's official borders. As suggested, we now address the above in our limitation section.

Contextual factors may also influence whether ethnic and national groups self-identify as a minority or a majority. For example, in the Israeli–Palestinian context, scholars have proposed distinguishing the national minority within Israel from the regional majority in the Middle East, a framing that may affect how the two groups perceive one another [104].

104. Reiter, Yitzhak. National minority, regional majority: Palestinian Arabs versus Jews in Israel. Syracuse University Press, 2009.

Reviewer #2

1. I did not find any background on the Israeli–Palestinian conflict. Providing some context on this issue would help readers unfamiliar with it better understand the importance of the study.

Our response:

As recommended, we added a brief factual background on the Israeli–Palestinian conflict. We keep this overview minimal and strictly contextual, as a comprehensive history is outside the manuscript’s scope. While we agree with the reviewer that further information on the Israeli-Palestinian conflict may assist the readers appreciate the implications of our work, we believe that our results are not limited to the specific situation in which the experiments were conducted.

2. In the manuscript, you refer to the Arab participants as “Israeli Arabs.” Did you ask the participants about their self-identification? In many publications this group is referred to as “Arabs–Palestinians who live in Israel,” so it may be worth clarifying or justifying your choice of terminology.

Our response:

Given that our experimental manipulation relied on identity priming, we avoided from requesting participants to define their identities. We now add a clarification of our choice of terminology as well as our sample criteria in ’The current study’ section.

3. It would also be helpful to include a clear definition of how the majority of this group self-identifies, so that readers understand whom the term refers to.

Our response:

We add as suggested, clarification in ’The current study’ section.

The demographic profile inside Israel's boundaries stabilized only after the 1948 war and the 1949 ceasefire agreements with the Arab states. Although Arabs became citizens after May 1948 and are legally incorporated into the Israeli state, they are excluded from the Jewish national belonging community due to Israel's self-identification as a Jewish state and home of the Jewish people. This state of affairs can make it extremely challenging for majority and minority group members alike to imagine themselves in each other’s position [54]. Furthermore, the national and ethnic identification in this context is a topic of controversy and has broad political ramifications. Specifically, studies on the Arab minority in Israel that dealt with the issue of identity were unable to reach an agreement and proposed multiple names as well as sub-groups divisions for this population (e.g. Arabs–Palestinians, Bedouin, Druses) [55]. Despite the foregoing, one of the distinctive characteristics of the whole Arab population of the State of Israel is their mother tongue, Arabic, which is recognized as the official language for the minority in Israel [56]. According to the Israeli Central Bureau of Statistics (CBS), 82% of Jews use Hebrew as their primary language at home, whereas 96% of Arabs speak Arabic [57]. As a result, we employ native language as our sample criteria, that is Hebrew native language for majority group (Jews) and Arabic native language for minority group (Arabs).

54. Bar-Tal, D., & Teichman, Y. Stereotypes and Prejudice in Conflict: Representations of Arabs in Israeli Jewish Society. Cambridge: Cambridge University Press; 2005.

55. Hitman G. May 2021 Riots by the Arab minority in Israel: national, civil or religious? Contemporary Review of the Middle East. 2023 Oct 19;10(4):346–63. doi.org/10.1177/23477989231198326

56. Harel-Shalev, A. Arabic as a minority language in Israel: A comparative perspective. Adalah's Newsletter, 2005, 14: 1-10. Available from: https://www.adalah.org/uploads/oldfiles/newsletter/eng/jun05/ar1.pdf

57. Admon, T., Annual Topic – Use of Languages. In: Central Bureau of Statistics 2021 (Israel) .הסקר החברתי: נושא שנתי – שימוש בשפות. [Hebrew]. Jerusalem: Central Bureau of Statistics; 2024 March. Available from: https://www.cbs.gov.il/he/publications/DocLib/2024/1905/h_print.pdf

---

## [Editor Report · Decision Letter 1]

25 Nov 2025

If I were you

Minority and majority members evaluate relevancy and subjective experience differently while putting themselves in the other's shoes

PONE-D-25-43032R1

Dear Dr. Milshtein,

We’re pleased to inform you that your manuscript has been judged scientifically suitable for publication and will be formally accepted for publication once it meets all outstanding technical requirements.

Kind regards,

Gal Harpaz, Ph.D.

Academic Editor

PLOS ONE

Additional Editor Comments (optional):

Dear Authors,

Thank you for your revised submission and for the thoughtful work you have put into strengthening your manuscript. After receiving the reviewers’ evaluations, I am pleased to inform you that your paper has been accepted for publication.

With best regards,

Dr. Gal Harpaz
---

## [Editor Report · Acceptance letter]

PONE-D-25-43032R1

PLOS One

Dear Dr. Milshtein,

I'm pleased to inform you that your manuscript has been deemed suitable for publication in PLOS One. Congratulations! Your manuscript is now being handed over to our production team.

Kind regards,

on behalf of

Dr. Gal Harpaz

Academic Editor

PLOS One